# Beyond Co-occurence: A Study of Early-stage Semantic Geometry in Next-Token Prediction

**Yize Zhao, Isabel Papadimitriou & Christos Thrampoulidis**
Department of Electrical and Computer Engineering
The University of British Columbia
`{zhaoyize, cthrampo}@ece.ubc.ca`
`isabel.papadimitriou@ubc.ca`

## Abstract

Neural Collapse predicts that balanced one-hot classification pushes model representations to be equally far from each other; a symmetric configuration that ignores any semantic similarity in the inputs. This creates a puzzle: next-token prediction language models are trained predominantly (as context length increases) with one-hot labels, yet they clearly learn that "red" and "blue" are more similar than "red" and "circle." How does gradient descent find such semantic structure when co-occurrence statistics collapse to one-hot sparsity, eliminating any shared next-tokens among different contexts? To investigate this tension we identify a controlled setting where inputs have latent semantic factors but are mapped to distinct one-hot labels. We find that semantic geometry emerges early in training: representations cluster by shared attributes despite receiving no explicit supervision to do so. This structure is transient: with sufficient capacity and time, the model eventually reaches the predicted symmetric state where all representations are equally separated. We study this phase transition through Gram matrix analysis and propose a preliminary modification to the commonly used unconstrained features model to capture the emerged semantic geometry.

## 1 Introduction

The emergence of semantic meaning from discrete tokens is naturally encoded in the geometric organization of the embedding space (BDVJ03). Classical frameworks model this geometry as the outcome of an implicit factorization of word-word co-occurrence statistics (LG14). Concretely, this shows that word embedding model like Word2vec (MSC$^+$13) and GloVe (PSG18) yield embeddings capturing latent relationships by implicitly reconstructing co-occurence patterns in the form of Pointwise Mutual Information (PMI) matrices. This paradigm, however, assumes a "dense" regime of local windows where token pairs frequently overlap, resulting in non-sparse co-occurence matrices.

In contrast, modern language models operate on a context-next-token structure, which is characterized by sparsity. Concretely, the conditional support for any given context is *sparse*—only a small number of tokens can realistically follow a specific context/sequence. Recent theoretical work has addressed this by modeling next-token-prediction training as supervision over *sparse* soft labels (ZBVT24; ZT25). For example, the label vector associated with the context "Brazil is famous for its" includes non-zero entries (reflecting empirical occurrences) for tokens "soccer", "coffee", and "carnival" and zeros for tokens "skiing", "desserts". These analyses demonstrate that Gradient Descent can recover a meaningful and semantically rich geometry under such sparse soft-labels, provided that these soft-label distributions act as a bridge to align representations of different contexts. While, compared to prior classical work, this analysis tackles the inherent sparsity in the conditional distribution of next-tokens, it still relies on the presence of shared label distributions among contexts to yield meaningful semantic representations.

However, this theoretical bridge collapses in the limit of modern long-context training. As the sequence length grows, the probability of encountering repeated contexts vanishes, forcing the training objective into a strictly one-hot regime. Here, no shared target label distribution exists to link disparate contexts. Despite that and despite the lack of any other explicit supervision

for similarity between contexts, large language models successfully recover complex semantic relationships ([BMR$^+$20]; [Eth19]).

From a representation-geometry viewpoint, this observation creates a fundamental tension with current theory: For one-hot supervision, the literature on Neural Collapse ($\mathcal{NC}$) ([PHD20]) predicts that the model's learnt representations should form a symmetric Simplex Equiangular Tight Frame (ETF) in which representations are pushed toward maximal separation. In this terminal state, the geometry is dictated solely by the one-hot labels and retains no trace of the underlying input semantics; all representations are maximally separated from each other. This presents a paradox:

*How can Gradient Descent find structured semantic geometry when the data is strictly one-hot?*
*If the terminal state of optimization is a symmetric ETF that erases latent relationships,*
*under what conditions does a semantic geometry emerge?*

To study this, we investigate whether the optimization process exhibits an implicit bias toward a structured semantic geometry during *early* training stages. To make this concrete, we characterize the Gram matrices of the context embeddings during training to identify the sufficient summary statistics of the input data that predict geometric alignment.

Our contributions are summarized as follows:

1. We identify a synthetic setting where Gradient Descent recovers structured, semantic geometry, despite purely one-hot next-token supervision.

2. Through detailed numerical analysis, we show that this semantic geometry is a transient phase occurring early in training; given sufficient model capacity and training time, it eventually yields its place to a symmetric ETF, reconciling our findings with standard $\mathcal{NC}$ theory.

3. We propose a simplified mathematical model that successfully reproduces this early-stage semantic alignment, offering a foundation for future theoretical work.

## 2  BACKGROUND

### 2.1  NEURAL COLLAPSE ($\mathcal{NC}$)

Neural Collapse characterizes the geometry of last-layer representations of a deep neural network trained on balanced one-hot labeled data in the Terminal Phase of Training (TPT), where training error is driven toward zero. Concretely, ([PHD20]) found that the penultimate layer representations (and classifier weights) simplify into a symmetric structure described as follows.

Let $H = [h_{1,1}, \ldots, h_{K,n_K}] \in \mathbb{R}^{d \times \sum n_k}$ be the feature matrix (last-layer embeddings) of examples belonging to $K$ classes, such that there is equal number of examples $n_1 = \ldots = n_K$ per class. Let $W = [w_1, \ldots, w_K]^\top \in \mathbb{R}^{K \times d}$ be the classifier weight matrix. Let $\mu_k = \frac{1}{n_k} \sum_{i=1}^{n_k} h_{k,i}$ be the $k$-th class mean and $\mu_G = \frac{1}{\sum n_k} \sum_{k,i} h_{k,i}$ be the global mean.

$\mathcal{NC}1$ (Variability Collapse): The within-class covariance $\Sigma_W$ vanishes. Metric$_{\mathcal{NC}1} = \Sigma_W \to 0$ where $\Sigma_W = \frac{1}{\sum n_k} \sum_{k=1}^{K} \sum_{i=1}^{n_k} (h_{k,i} - \mu_k)(h_{k,i} - \mu_k)^\top$.

$\mathcal{NC}2$ (Convergence to ETF): The centered class means $\tilde{H} = [\mu_1 - \mu_G, \ldots, \mu_K - \mu_G]$ converge to a Simplex Equiangular Tight Frame (ETF).

$$\text{Metric}_{\mathcal{NC}2} = \left\| \frac{\tilde{H}^\top \tilde{H}}{\|\tilde{H}^\top \tilde{H}\|_F} - \frac{1}{\sqrt{K-1}} \left( I_K - \frac{1}{K} 1_K 1_K^\top \right) \right\|_F \to 0$$

NC2 predicts that the representations of examples belonging to different classes are equally (in fact, maximally) separated. Here and throughout, we assume $d \geq K$. This assumption guarantees the existence of an ETF and it is required for NC. It is also used in ([ZBVT24]; [ZT25]) who show rich geometric arrangements emerge under soft-label information. Here, focusing on one-hot labels while maintaining $d \geq K$, allows us to study whether a rich geometry can emerge before the representation eventually yields an ETF.

## 2.2 The Unconstrained Feature Model (UFM)

The UFM is a theoretical abstraction used to study NC (MPP20). In this model, the penultimate layer features $h_{k,i}$ are treated as free optimization variables rather than the output of a specific architecture. For $\mathcal{L}$ the CE loss, the training objective is: $\min_{W,H} \sum_{k=1}^{K} \sum_{i=1}^{n_k} \mathcal{L}(W h_{k,i}, y_k)$. For balanced one-hot classification tasks, the UFM identifies the Simplex ETF as the unique global minimizer (JLZ+21; TKVB22). This provides a theoretical baseline for the "collapsed" terminal state; however, because the UFM treats features as free optimization variables, it abstracts away the input data. This abstraction precludes any analysis of how specific input features influence the resulting representations. Our study moves beyond this terminal analysis to investigate the emergence of semantic geometry during the training process. Specifically, we aim to characterize the relationship between the embedding structure and the underlying statistical properties of the input, a connection that the UFM is unable to capture.

## 3 The Toy One-Hot Data Model

We establish a controlled setting where diverse input contexts share latent structures yet are mapped to one-hot targets. This framework allows us to observe whether GD favors a geometry that reflects latent "semantic classes" or one that merely satisfies the maximal separation required by the loss function.

### 3.1 Latent Factors and Semantic Classes

We establish a controlled setting to investigate whether Gradient Descent favors a geometry reflecting latent "semantic classes" or one satisfying the maximal separation required by the loss function.

**Latent Factors.** We define the data generative process through independent factors $F$, each consisting of a discrete set of attributes $\mathcal{A}_F$. Specifically, we define a Color Factor ($F = \mathcal{C}$) containing attributes $\mathcal{A}_\mathcal{C} = \{\text{RED}, \text{BLUE}, \dots\}$ and a Shape Factor ($F = \mathcal{H}$) containing attributes $\mathcal{A}_\mathcal{H} = \{\text{CIRCLE}, \text{SQUARE}, \dots\}$.

**Semantic Class.** We define a Semantic Class $\mathcal{S}_a$ as the set of all samples sharing a specific attribute $a \in \mathcal{A}_F$. For example, the semantic class $\mathcal{S}_{\text{RED}}$ contains all samples generated with the "Red" attribute, regardless of their shape. These classes are not mutually exclusive; every sample belongs to exactly one Color class and one Shape class simultaneously.

**Label Class.** The optimization objective is defined by the Label Class $\mathcal{Y}$. A label class $y_k$ corresponds to the unique intersection of semantic factors, $y_k = (a_c, a_h)$, resulting in $K = |\mathcal{A}_\mathcal{C}| \times |\mathcal{A}_\mathcal{H}|$ mutually exclusive one-hot targets.

**Context Samples.** To ensure the model infers structure from statistics rather than memorization, we introduce variability within each class. Each input is a triplet $x = [t_{\text{ID}}, t_c, t_h]$. Here, $t_{\text{ID}}$ is a nuisance token perfectly predictive of $y_k$ but sharing no information across classes. The contextual tokens $t_c, t_h$ are drawn from sets of fine-grained variants $\mathcal{T}_a$ specific to an attribute $a$ (e.g., when $a = \text{RED}$, the variants can be $\{\text{DARK\_RED}, \text{LIGHT\_RED}, \text{MEDIUM\_RED}\}$). Because the model never observes the coarse latent factors directly, it must infer the semantic information $\mathcal{S}$ from the co-occurrence statistics of these fine-grained tokens.

### 3.2 Competing Geometric Regimes

The interplay between the generative process and the optimization objective establishes a fundamental tension. We characterize this as a competition between two distinct pressures, each driving the embeddings toward a specific geometric configuration:

**1. Semantic Regime:** Since attributes (e.g., "Red") are the common driver for multiple targets, there is a pressure to encode this shared factor. Consequently, the model maps diverse fine-grained tokens (e.g., DARK\_RED, LIGHT\_RED) into a tight semantic cluster, preserving the latent similarity between distinct label classes despite possibly different training target label classes.

**2. Collapse Regime:** Conversely, the one-hot loss exerts a pressure to maximize the margin between distinct label classes. Geometrically, this drives the representation toward a symmetric Simplex ETF,

where all labels are mutually equidistant and orthogonal, effectively erasing the shared semantic structure.

### 3.3 QUANTIFYING SEMANTIC GEOMETRY

To detect semantic structure, we adapt Neural Collapse metrics to evaluate geometry with respect to latent attributes rather than label classes.

**Semantic Convergence ($\mathcal{NC}1_{\mathbf{sem}}$).** We compute the standard $\mathcal{NC}1$ metric from the classical literature, but with a critical modification: instead of grouping samples by their label class $y_k$, we group them by their semantic class (attribute) $a \in \mathcal{A}_F$. Let $\mu^{(a)}$ denote the centroid of the semantic class $\mathcal{S}_a$ and $\mu_G^{(F)}$ denote the global mean over all embeddings from attributes in $F$. We define the within-class ($\Sigma_W$) and between-class ($\Sigma_B$) scatter matrices as: $\Sigma_W(F) = \frac{1}{N} \sum_{a \in \mathcal{A}_F} \sum h \in \mathcal{S}_a (h - \mu^{(a)})(h - \mu^{(a)})^\top$ and $\Sigma_B(F) = \frac{1}{|\mathcal{A}_F|} \sum_{a \in \mathcal{A}_F} (\mu^{(a)} - \mu_G^{(F)})(\mu^{(a)} - \mu_G^{(F)})^\top$.

We quantify the collapse by the trace ratio $\mathcal{NC}1_{sem}(F) = \text{Tr}(\Sigma_W)/\text{Tr}(\Sigma_B)$. A lower score indicates that sample embeddings with shared attributes have clustered tightly around a common centroid.

**The Semantic Alignment Ratio ($\mathcal{NC}1_{\mathbf{sem}}$).** We normalize our observation against the expected terminal value of the ETF baseline:

$$\mathcal{R}_{\text{sem}}(F) = \frac{\mathcal{NC}1_{\text{sem}}(F)}{\mathcal{NC}1_{\text{ETF}}(F)} \tag{1}$$

This ratio serves as our primary order parameter. $\mathcal{R}_{\text{sem}} < 1.0$ identifies the Semantic Regime, where geometry is more structured than the UFM baseline. $\mathcal{R}_{\text{sem}} \approx 1.0$ indicates the Collapsed Regime, where geometry is dictated by label class. We also validate these regimes qualitatively by inspecting the Gram matrix $G = H^\top H$ with rows/columns sorted by attribute.

## 4 EXPERIMENTS

### 4.1 EXPERIMENTAL SETUP

**Data Specification.** We generate a dataset with two latent factors: Color ($|\mathcal{A}_\mathcal{C}| = 2$) and Shape ($|\mathcal{A}_\mathcal{H}| = 4$), resulting in $K = 8$ label classes. The input vocabulary ($V = 818$) comprises 800 unique Nuisance IDs (100 per label class) and 18 Contextual Tokens (3 fine-grained variants for each of the 6 attributes). The total dataset consists of $N = 800$ samples.

**Model Architecture.** We train a standard Transformer decoder without positional encodings. To test capacity dependence, we sweep depth $L \in \{1, 2, 4, 8\}$ and embedding dimension $d \in \{8, 16, 32, 64\}$. The minimum dimension ($d = 8$) satisfies $d \geq K$, ensuring the representation space is not rank-constrained; any collapse is strictly a product of optimization dynamics, not architectural bottlenecks. All models utilize $H = 4$ attention heads.

**Optimization.** Training is performed with full-batch Adam (lr=$5 \times 10^{-4}$) for $10^4$ epochs.

### 4.2 THE TRANSIENT SEMANTIC PHASE

We track the Semantic Alignment Ratio $\mathcal{R}_{\text{sem}}$ to characterize representation dynamics. Figure 1 reveals that the optimization trajectory is non-monotonic. In the early regime, $\mathcal{R}_{sem}$ drops significantly below the ETF baseline (1.0), confirming that embeddings cluster around shared attributes (e.g., "Red"). Geometric inspection (Figure 1, Top) validates this: at peak semantics, the Gram matrix displays a clear similarity hierarchy based on attribute overlap. We observe strong alignment (dark red) between samples sharing both attributes and moderate alignment (lighter red) for partial matches sharing only one attribute (e.g., same Color, different Shape). In contrast, pairs with no overlapping attributes are anticorrelated (blue). However, at the terminal stage, the pressure for label separation dominates. The semantic structure disappears, and the geometry converges to the rigid, block-diagonal Simplex ETF.

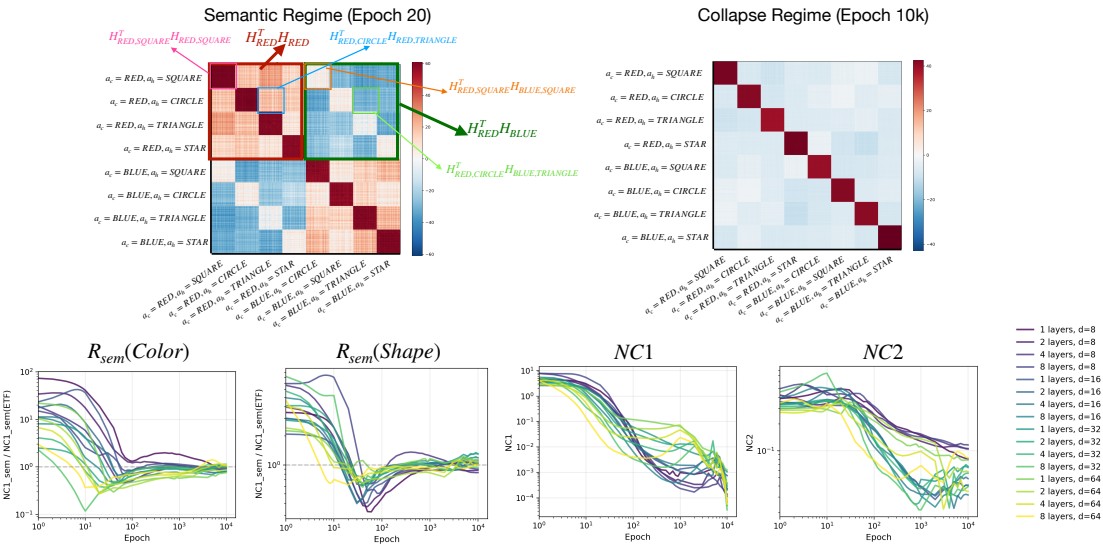

Figure 1: **The Transient Semantic Phase and Terminal Collapse. (Top) Geometric Visualization.**
Gram matrices ($G = H^\top H$) with rows/columns sorted by semantic attribute. **Left (Semantic Regime):**
At Epoch 20, the geometry exhibits a semantic structure. The coarse intra-attribute block $H_{\text{RED}}^\top H_{\text{RED}}$
(highlighted in red) separates from the inter-attribute block $H_{\text{RED}}^\top H_{\text{BLUE}}$ (highlighted in dark green).
Crucially, strong off-diagonal energy exists even between distinct labels (e.g., $H_{\text{RED,CIRCLE}}^\top H_{\text{RED,TRIANGLE}}$),
confirming that the model has learned the shared "Red" attribute. **Right (Collapse Regime):** At Epoch 10k,
this semantic structure is erased. The matrix converges to the rigid Simplex ETF, where all off-diagonal
correlations vanish and only the class-specific blocks (e.g., $H_{\text{RED,SQUARE}}^\top H_{\text{RED,SQUARE}}$) remain. **(Bottom)
Geometry Dynamics.** $\mathcal{R}_{sem}$ **(Left Pair):** The "U-shaped" trajectory confirms the transient nature of the
semantic phase. **NC1 / NC2 (Right Pair):** Standard collapse metrics decay to zero, verifying that all
models converge to ETF at the terminal stage.

## 5 THEORETICAL PROXY FOR SEMANTIC REGIME

To isolate the mechanism driving the semantic phase, we introduce a simplified "bag-of-words"
model. We define the network logit $\mathbf{z}_i = WHx_i$. Where $x_i$ is a multi-hot vector has length of
input vocabulary size, We train the model on label class with cross entropy. This model isolates
the semantic regime. Unlike the deep Transformer, it does not exhibit terminal Neural Collapse.
Instead, it converges specifically to the hierarchical semantic geometry observed in our experiments.
In Appendix A, we show that this stable solution arises from the joint diagonalization of input and
label correlation matrices, providing a theoretical basis for semantic geometry analysis.

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

## A    Theoretical Analysis of the Log-Linear Proxy

To theoretically ground the transient semantic alignment observed in the main text, we analyze a simplified log-linear model. This model isolates the linear interaction between token co-occurrence (input geometry) and label structure (task geometry), removing the non-linear complexities of the deep Transformer.

### A.1    Model Formulation

Consider a dataset of $n$ contexts. For the $i$-th context, let $\mathcal{V}(i) \subseteq \mathcal{V}$ denote the set of constituent token IDs. We represent each context as a multi-hot vector $x_i \in \{0,1\}^V$, where:

$$(x_i)_v = \mathbb{I}\{v \in \mathcal{V}(i)\}.$$

We assume a bag-of-words representation where token position is ignored (absorbing a normalization factor of $1/L$ into the learning rate if necessary). The label for context $i$ is $y_i \in [K]$. The model parametrizes the mapping using a rank-$d$ factorization $Z = WH$, where $H \in \mathbb{R}^{d \times V}$ contains token embeddings and $W \in \mathbb{R}^{K \times d}$ is the umemdedding matrix. The logits are given by $z_i = WHx_i = Zx_i$. The network minimizes the cross-entropy loss:

$$\mathcal{L}(W, H) = -\sum_{i=1}^{n} \log \sigma(z_i)_{y_i},$$

where $\sigma(\cdot)$ is the softmax function.

### A.2    Gradient Dynamics

The gradients with respect to the factors capture the competition between updating the readout $W$ (based on current embeddings) and updating the embeddings $H$ (based on the current readout).

$$\nabla_W \mathcal{L} = \sum_{i=1}^{n} (\sigma(zi) - ey_i) xi^\top H^\top, \ \nabla_H \mathcal{L} = W^\top \sum_{i=1}^{n} (\sigma(z_i) - ey_i) x_i^\top. \tag{2}$$

Looking specifically at a single token $v \in \mathcal{V}$ (column $h_v$ of $H$), the update aggregates error signals from all contexts $\mathcal{I}(v)$ where the token appears:

$$\nabla_{h_v} \mathcal{L} = \sum_{i \in \mathcal{I}(v)} W^\top (\sigma(z_i) - e_{y_i}).$$

This explicitly shows that a token's embedding is pushed by the average gradient of the classes it co-occurs with. Semantic tokens (which co-occur with multiple classes) receive mixed signals, whereas class-exclusive tokens receive pure signals.

### A.3    Separability and Convergence

The loss can be driven to zero if and only if there exists a matrix $Z \in \mathbb{R}^{K \times V}$ with rank$(Z) \leq d$ such that the data is linearly separable:

$$\forall i \in [n], \forall u \neq y_i : \quad (e_{y_i} - e_u)^\top Z x_i > 0.$$

Provided $d$ is sufficiently large (e.g., $d \geq$ rank$(Z^*)$ for a separable solution $Z^*$), a feasible factorization always exists. For our synthetic dataset, separability is guaranteed by the presence of nuisance tokens.

**Claim 1** (Sufficient Condition for Separability). *The dataset is separable if, for every context $i$, there exists at least one constituent token $v \in \mathcal{V}(i)$ that is **class-exclusive**. That is, $v$ appears only in contexts belonging to class $y_i$.*

*Proof.* For each class $c$, let $\mathcal{U}(c)$ be the set of tokens unique to that class. By assumption, $\mathcal{U}(c) \cap \mathcal{V}(i) \neq \emptyset$ for all $i$ with $y_i = c$. We construct a solution $Z$ explicitly:

$$Z_{c,v} = \mathbb{I}\{v \in \mathcal{U}(c)\}, \quad Z_{c',v} = 0 \text{ for } c' \neq c.$$

For any input $i$ with label $y_i$ and any incorrect class $c \neq y_i$:

$$(e_{y_i} - e_c)^\top Z x_i = \sum_{v \in \mathcal{V}(i)} (\mathbb{I}\{v \in \mathcal{U}(y_i)\} - 0) \geq 1.$$

Since the margin is positive, the loss can be driven to zero. $\qquad\qquad\qquad\qquad\qquad\qquad$ □

Under these separability conditions, it is known that gradient descent on $W, H$ induces an implicit regularization, converging directionally to the solution of the nuclear-norm minimization problem:

$$\min_Z ||Z||_* \quad \text{s.t. } (ey_i - e_u)^\top Z x_i \geq 1 \quad \forall i, u \neq y_i. \tag{3}$$

This suggests the model prefers low-rank solutions that explain the data using the fewest dimensions—initially exploiting the shared semantic factors before eventually overfitting to the high-rank nuisance IDs.

## A.4   Joint Diagonalizability Condition

To understand the geometry of the learned features, we analyze the relationship between the input correlation structure and the label structure. Let $X \in \mathbb{R}^{V \times n}$ be the matrix of input vectors (columns $x_i$) and $Y \in \mathbb{R}^{K \times n}$ be the one-hot label matrix. In a balanced setup, we can model the label interaction matrix as $M = Y^\top Y = I_K \otimes 1_{n/k} 1_{n/k}^\top$. The question of whether the model learns "semantic" features reduces to a geometric question: Can the singular value decomposition (SVD) of the input-output cross-covariance $YX^\top$ share the right singular vectors with the input covariance $XX^\top$? Let $XX^\top = VDV^\top$ be the eigendecomposition of the input correlations. For $YX^\top$ to admit a factorization $USV^\top$ (sharing the same basis $V$), the matrix $V$ must simultaneously diagonalize $XX^\top$ and the gram matrix of the cross-covariance:

$$(YX^\top)^\top(YX^\top) = XY^\top YX^\top = XMX^\top.$$

This simultaneous diagonalization is possible if and only if the two symmetric matrices commute:

$$(XX^\top)(XMX^\top) = (XMX^\top)(XX^\top).$$

When this commutativity holds, the principal directions of the input data (eigenvectors of $XX^\top$) align perfectly with the discriminative directions required by the task. In our dataset, the semantic tokens (Color/Shape) form strong blocks in $XX^\top$ that naturally commute with the class block structure of $M$, explaining why the model aligns with these semantic axes in the early training regime. In particular, this joint diagonalization property can allow us to utilize the spectral-initialization framework of (SMG13; SMG19) to formulate the semantic geometry.

