# OpenReview forum: "Beyond Co-occurence: A Study of Early-stage Semantic Geometry in Next-Token Prediction"
_ICLR.cc/2026/Workshop/GRaM — ICLR 2026 Workshop GRaM Poster_

### Official Review · Reviewer_YYvK · 2026-02-21
**An interesting transient semantic phase in LM training: quantifiable Insights but limited by synthetic data and simplified assumptions**

**Rating:** 6
**Confidence:** 3

**Review:**

This paper identifies a transient semantic phase that occurs early in training language models. Using a controlled synthetic dataset with latent factors like "Color" and "Shape," the researchers show that representations cluster by shared attributes before eventually succumbing to the symmetric, feature-erasing terminal state of optimization. They quantify this transition using the Semantic Alignment Ratio, which serves as an order parameter to distinguish between semantic and collapsed regimes.

Strengths:
1. This paper provides a temporal explanation for how models can possess ''meaningful" embeddings while still moving toward a ''collapsed" terminal state, bridging the gap between classical co-occurrence theory and modern Neural Collapse.
2. Through capacity sweeps, the authors discover that moderate embedding dimensions produce the deepest semantic alignment, whereas high-capacity models often ''shortcut" the semantic phase to reach collapse faster.
3. The introduction of the $R_{sem}$ ratio and the use of Gram matrix visualizations provide a robust, quantifiable way to track geometric evolution beyond qualitative observations.

Weaknesses:
1. The experiments are conducted on a small scale, which may not represent the complex, heavy-tailed distributions of real-world language.
2. The theoretical log-linear proxy assumes a bag-of-words representation that ignores token positions, potentially missing how sequence structure interacts with the "transient phase" in actual Transformers.

**Pmlr Suitability:**

NA

---

### Official Review · Reviewer_263p · 2026-02-23
**An Interesting Framework for Understanding Semantic Emergence and the Path to Neural Collapse**

**Rating:** 7
**Confidence:** 2

**Review:**

### **Summary**

This paper investigates how language models learn semantic relationships (such as color similarity) despite being trained on one-hot targets that, in theory, push the model toward a non-semantic, symmetric state known as a Simplex Equiangular Tight Frame (ETF). Through a well-designed synthetic framework, the authors demonstrate that semantic geometry emerges as a transient phase during training before the final "Neural Collapse" takes over.

### **Strengths**

* **Interesting Starting Point:** The work serves as an excellent foundation for studying the tension between semantic emergence and Neural Collapse (NC). It addresses a fundamental question in NLP with a clear, focused lens.
* **Methodological Rigor:** The creation of a controlled synthetic dataset that allows for the measurement of both standard NC (ETF) and "Semantic" NC is a major contribution. This framework is extremely rich and significantly helps build a better intuition for how optimization dynamics operate in next-token prediction.
* **Controllability:** While the final results are intuitive, the paper’s primary value lies in the precise control over data and architecture, allowing the authors to isolate variables that are often "hidden" in large-scale LLM training.

### **Suggestions for Improvement & Questions**

* **Figure 1 Clarity:** I noticed that Figure 1 contains two distinct parts, but only one of them is discussed in detail within the text. To improve the flow, I would suggest keeping the most impactful visualization in the main body and moving the other to the appendix, or ensuring both are clearly analyzed.
* **Appendix B:** There is currently no formal citation or link to Appendix B in the main text. Please ensure you reference it to guide the reader.
* **Appendix C:** This section feels somewhat disconnected from the core narrative. I would advise the authors to either establish a stronger theoretical or empirical bridge to the main paper or to remove it to keep the submission focused. I believe the best option is to remove.


* **Exploring Constraints:** In Appendix B, I was curious about the effects of even more intense capacity constraints (e.g.,  or ). Seeing where the "semantic bridge" truly breaks under extreme constraints (d=8, d=12) would provide even deeper insights into what information is lost first. A full grid evaluation would be really rich.

### **Final Remark**

This is a high-quality submission that fits the scope of the GRaM workshop perfectly. By addressing minor editing issues with the figures and appendices, the paper will be even stronger.

**Pmlr Suitability:**

NA

---

### Meta-Review · Area_Chair_s217 · 2026-02-27

**Decision:**

Accept

**Metareview:**

The paper investigates the apparent paradox of how next token prediction models learn rich semantic representations, even when trained with sparse, one-hot labels. Frameworks such as neural collapse posit that such settings should result in a rigid state. A controlled synthetic dataset is designed where the inputs have fixed latent semantic factors, which are mapped to mutually exclusive one-hot targets. The study shows that semantic geometry might be a transient phase that only emerges in the training phase. A simplified log-linear proxy model is also proposed to understand how this early semantic alignment occurs. This is presented as a tiny paper. The reviewers have liked the paper and voted for accepting it. I also think that the paper could be a good tiny paper contribution.

**Relevance To Proceedings:**

Tiny paper — does not apply

**Relevance To Workshop:**

Yes — suitable for GRaM

---

### Decision · Program_Chairs · 2026-03-02

Accept (Poster)